# Decomposing Interventional Causality into Synergistic, Redundant, and Unique Components

**Abel Jansma**
Dutch Institute for Emergent Phenomena, the Netherlands
Institute for Logic, Language and Computation, University of Amsterdam
Institute of Physics, University of Amsterdam
`a.a.a.jansma@uva.nl`

## Abstract

We introduce a novel framework for decomposing interventional causal effects into synergistic, redundant, and unique components, building on the intuition of Partial Information Decomposition (PID) and the principle of Möbius inversion. While recent work has explored a similar decomposition of an observational measure, we argue that a proper causal decomposition must be interventional in nature. We develop a mathematical approach that systematically quantifies how causal power is distributed among variables in a system, using a recently derived closed-form expression for the Möbius function of the redundancy lattice. The formalism is then illustrated by decomposing the causal power in logic gates, cellular automata, chemical reaction networks, and a transformer language model. Our results reveal how the distribution of causal power can be context- and parameter-dependent. The decomposition provides new insights into complex systems by revealing how causal influences are shared and combined among multiple variables, with potential applications ranging from attribution of responsibility in legal or AI systems, to the analysis of biological networks or climate models.

## 1 Introduction

Causal language is ubiquitous throughout the natural and social sciences. A ball falls towards the earth *because* of a gravitational force, inflation rises *because* of excessive money supply, climate changes *because* of greenhouse gas emissions. In each case, we seek not just to describe what happens, but to understand why it happens—to identify the causal mechanisms underlying the observed phenomenon. Implicit in causal language are claims about the effect of interventions or counterfactual scenarios (we can mitigate climate change by reducing greenhouse gas emissions, and inflation would not be as high if there had been no excessive money supply). When we study something that is embedded in a web of complex causal interactions, attributing causality can become more difficult, so sophisticated mathematical and computational techniques have been developed that all rely on one of two things: either one is able to directly intervene in the system and study its response, or one requires *a priori* knowledge of the causal dependencies in the system. In this study, we focus on situations in which causal effects are identifiable, and address the question of causal attribution [20, 7]. Given that we know the effect of interventions on various variables, how do we attribute the causal power to the different variables? In particular, we are interested in distinguishing between three types of attribution: unique, redundant, or synergistic causal power. Decomposition into these three classes is commonly done in information theory, where it is known as the *Partial Information Decomposition* (PID, [25]). Given two coin flips, for example, the answer to the question 'was there an even number of heads' is carried purely synergistically by the pair of coins, since each coin individually carries no information whatsoever about the answer.

39th Conference on Neural Information Processing Systems (NeurIPS 2025).

Here we extend this approach to causal quantities and define a 'partial causality decomposition'. Disentangling these three different components of causality is crucial to a good understanding and control of complex systems. DiFrisco and Jaeger [4], for example, already emphasised the importance of identifying and distinguishing redundant and synergistic causality in gene regulatory systems. In machine learning it is common to obtain predictions from opaque models whose internal reasoning is not clear. Understanding and attributing causal power, blame, or responsibility to input features is crucial to a safe and reliable deployment of such models. In machine learning, this is commonly done using Shapley values [24, 22], which are closely related to the decomposition presented here [13], but fail to disentangle unique, redundant, and synergistic contributions.

Redundant and synergistic causality are related to the established notions of causal sufficiency and necessity [20, 9, 7]. If a set of variables carries redundant causal power, then any of the variables suffice to affect the outcome, whereas exerting the synergistic causal power of a set of variables necessitates a joint intervention. However, sufficiency and necessity are usually defined with respect to individual outcomes (rung 3 of Pearl's ladder of causation), whereas in this study we decompose an average causal effect (namely, the maximum average causal effect in Equation (8), which inhabits rung 2). A brief example of how one could apply the framework to study sufficient and necessary causes is included in Appendix C.

Recently, Martínez-Sánchez et al. [19] demonstrated a technique to decompose an observational measure (based on mutual information) that they refer to as causal. We, however, are of the opinion that a measure of causality should necessarily be interventional, and not accessible from the joint distribution alone [20]. Still, Martínez-Sánchez et al. [19] raise an interesting question, namely, is it possible to disentangle the unique, redundant, and synergistic causal power of a set of variables? We will show that this is indeed possible, drawing inspiration from the Partial Information Decomposition [25] and the use of Möbius inversions in complex systems [13].

## 2 Background

### 2.1 Partial orders and Möbius inversion

To show the algebraic structure of the decomposition, we first need to introduce a number of concepts. First are partially ordered sets, or *posets*:

**Definition 1** (Partially Ordered Set). *A partially ordered set is a tuple $(S, \leq)$, where $S$ is a set and $\leq$ is a binary relation on $S$ that is reflexive, antisymmetric, and transitive.*

When the ordering is clear from context, we sometimes use the shorthand $S$ to denote the poset. An example of a partial order is $(\mathcal{P}(T), \subseteq)$, the power set of $T$ ordered by inclusion. Given a poset $S$ and $a, b \in S$, the interval $[a, b]$ is the set $\{x : a \leq x \leq b\}$. If all intervals on $S$ are finite sets, then $S$ is called locally finite. We also define the following:

**Definition 2** ((Anti)chains). *Let $(S, \leq)$ be a poset. A subset $T \subseteq S$ is a chain if for all $a, b \in T$, $a \leq b$ or $b \leq a$. If for all $a, b \in T$, $a \leq b$ implies $a = b$, then $T$ is an antichain.*

Note that in particular any single element subset of a poset is both a chain and an antichain.

Functions on $S$ can interact with the partial order in various ways. One such function is the Möbius function:

**Definition 3** (Möbius function). *Let $(S, \leq)$ be a locally finite poset. Then the Möbius function $\mu_S : S \times S \to \mathbb{Z}$ is defined as*

$$\mu_S(x, y) = \begin{cases} 1 & \text{if } x = y \\ -\sum_{z : x \leq z < y} \mu_S(x, z) & \text{if } x < y \\ 0 & \text{otherwise} \end{cases} \tag{1}$$

This allows us to state the Möbius inversion theorem (MIT):

**Theorem 1** (Möbius inversion theorem, Rota [21]). *Let $(S, \leq)$ be a locally finite poset. Let $f, g : S \to \mathbb{R}$ be functions on $S$, and let $\mu_S$ be the Möbius function on $S$. Then*

$$f(b) = \sum_{a \leq b} g(a) \quad \Longleftrightarrow \quad g(b) = \sum_{a \leq b} \mu_S(a, b) f(a) \tag{2}$$

The Möbius inversion theorem states that sums of a function over a partial order can be inverted using the Möbius function. Because the l.h.s. of (2) can be interpreted as a discrete integral over the poset, the convolution with the Möbius function on the r.h.s. is often considered as a generalised discrete derivative. Indeed, applying the MIT to the natural numbers with their usual ordering recovers a discrete version of the fundamental theorem of calculus. As many quantities in complex systems correspond to integrals or derivatives over partial orders, the Möbius inversion theorem is a powerful tool in the analysis of such systems [13].

## 2.2 Decompositions into antichains

Principled decomposition into synergistic, redundant, and unique components was first explored by Williams and Beer [25] in the context of information theory under the name *Partial Information Decomposition* (PID). Their approach forms our main inspiration, so we briefly describe it here. Consider the mutual information $I(X_1, X_2; Y)$ that two variables $(X_1, X_2)$ carry about a variable $Y$. If $I(X_1, X_2; Y) = v$, then one might ask: *How* are the $v$ bits about $Y$ carried by the pair $(X_1, X_2)$? One could imagine that both variables carry the same $v$ bits, so that the information is carried redundantly, or that both variables uniquely carry $\frac{v}{2}$. Another option is that neither variable has information by itself, but it is the joint state of the pair that carries the $v$ bits synergistically. For two variables, one therefore writes

$$I(X_1, X_2; Y) = I_\partial(\{X_1\}; Y) + I_\partial(\{X_2\}; Y)$$
$$+ I_\partial(\{X_1, X_2\}; Y) + I_\partial(\{\{X_1\}, \{X_2\}\}; Y) \tag{3}$$

where the 'partial' information $I_\partial(\{X_i\}; Y)$ is the information that variable $X_i$ *uniquely* carries about $Y$, $I_\partial(\{\{X_1\}, \{X_2\}\}; Y)$ is the information that is carried *redundantly* by the two variables, and $I_\partial(\{X_1, X_2\}; Y)$ is the information that is carried *synergistically* by the joint state of the pair. Note, however, that the situation becomes more complex if more variables are added. Three variables $\{X_1, X_2, X_3\}$, for example, can carry information redundantly between the joint state of $\{X_1, X_2\}$ and the state of $\{X_3\}$, or synergistically between all three variables, etc. The central insight of the PID is that information can be carried redundantly among all incomparable subsets of variables. Given a set of variables $S$, the incomparable subsets of $S$ are the *antichains* of $(\mathcal{P}(S), \subseteq)$, denoted $\mathcal{A}(S)$. Some elements of $\mathcal{A}(\{X_1, X_2, X_3\})$ and their interpretation are:

- $\{\{X_1, X_2, X_3\}\} \rightarrow$ synergy among $X_1, X_2, X_3$
- $\{\{X_1, X_2\}, \{X_2, X_3\}\} \rightarrow$ redundancy between $\{X_1, X_2\}$ and $\{X_2, X_3\}$
- $\{\{X_1\}, \{X_2\}, \{X_3\}\} \rightarrow$ redundancy between $X_1, X_2, X_3$

We sometimes use simplified notation of the type $\{\{X_1, X_2\}, \{X_2, X_3\}\} = X_1 X_2 | X_2 X_3$. The set $\{\{X_1, X_2\}, \{X_1, X_2, X_3\}\}$ is *not* an antichain, because $\{X_1, X_2\} \subseteq \{X_1, X_2, X_3\}$. Terms like this should be excluded because the redundancy between $\{X_1, X_2\}$ and $\{X_1, X_2, X_3\}$ is simply the contribution of $\{\{X_1, X_2\}\}$. We will usually denote sets with uppercase Latin letters, and antichains with lowercase Greek letters.

A nice property of antichains is that they can be ordered with respect to redundancy by setting $\alpha \leq \beta$ when it is at least as 'easy' to access the information in $\alpha$ as that in $\beta$. For example, we set $\{\{X_1, X_2\}, \{X_3\}\} \leq \{\{X_1, X_2\}, \{X_3, X_4\}\}$, because you can access the information in $\{\{X_1, X_2\}, \{X_3\}\}$ by observing the pair $(X_1, X_2)$ or the single variable $X_3$, whereas accessing the information in $\{\{X_1, X_2\}, \{X_3, X_4\}\}$ necessarily requires observing a pair. This ordering can be formally defined as follows:

$$\alpha \leq \beta \quad \Longleftrightarrow \quad \forall B \in \beta, \exists A \in \alpha : A \subseteq B \tag{4}$$

The redundancy ordering is shown for $n = 2, 3, 4$ in Figure 1, and allows us to write the decomposition into antichains as

$$I(X; Y) = \sum_{\alpha \in \mathcal{A}(X): \alpha \leq X} I_\partial(\alpha; Y) \tag{5}$$

which can then be inverted using the MIT to find the contribution of each of the types of information:

$$I_\partial(\beta; Y) = \sum_{\alpha \leq \beta} \mu_{\mathcal{A}(X)}(\alpha, \beta) I_\cap(\alpha; Y) \tag{6}$$

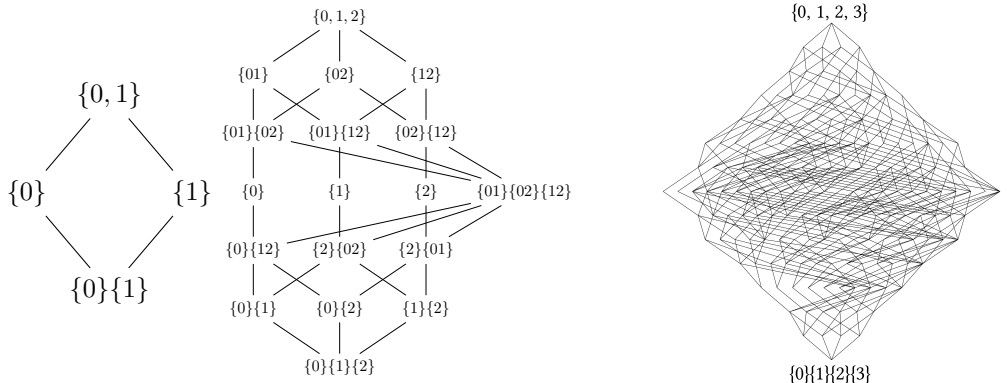

Figure 1: The transitive reduction (Hasse diagram) of the redundancy ordering from Equation (4) for the $n = 2, 3, 4$ variables. The posets contain respectively 4, 18, and 166 elements.

On the r.h.s. of Equation (6), we added a subscript to the mutual information $I_\cap(\alpha; Y)$, since it is now evaluated both on sets of variables, but also on other antichains. On a set $S$ of variables, $I_\cap(S; Y) = I(S, Y)$, but on a general antichain $\alpha$, $I_\cap(\alpha; Y)$ is the information that the variables in $\alpha$ share about $Y$ ($\cap$ referring to 'sharedness'). There are many ways to define this, and a significant portion of the PID literature has focused on exploring different approaches to fix the definition of $I_\cap$ (see e.g. [2, 6, 3, 10, 11, 5]). For example, Williams and Beer [25] introduced the PID with the measure $I_{min}(\alpha; Y) = \sum_y p(y) \min_{A \in \alpha} I(Y = y; A)$, that quantifies the expected "specific" information that any source $A \in \alpha$ provides about outcomes of $Y$. Since this measure was found to behave unintuitively on some distributions, a common alternative is the minimum mutual information (MMI) $I_{mmi}(\alpha; Y) = \min_{A \in \alpha} I(Y, A)$ [1].

A second complication in performing an antichain decomposition is that the number of ways information can be carried among $n$ variables grows superexponentially with $n$ (namely, as the $n$th Dedekind number). This makes recursively solving the system of equations in (5), or recursively calculating the Möbius function in (6), computationally very expensive. This has limited the PID approach mostly to systems with $n \leq 3$. Recently, however, a closed-form expression for the Möbius function on $\mathcal{A}(S)$ for any $S$ was derived, which can offer a double-exponential speedup and opens up the possibility to study larger systems [14]. However, to avoid the complexity of the computation obscuring the simplicity of the method, we chose here to focus on decomposing quantities of up to three variables, for which such speedups are not necessary.

## 3 Interventional causality

### 3.1 Average causal effects

A central object of study in causality is the average treatment effect [20]. Given a set of variables $X_S := \{X_i | i \in S\}$, where $S$ is some indexing set, the average treatment effect (ATE) of $X_S$ on an outcome $Y$ is defined as

$$\text{ATE}(X_S; Y) = \mathbb{E}[Y | do(X_S = x_1)] - \mathbb{E}[Y | do(X_S = x_0)] \tag{7}$$

where the do-operator denotes an intervention, and $x_1$ and $x_0$ denote the 'treated' and 'untreated' state, respectively. The most direct way to estimate quantities like (7) is to perform a randomised controlled trial, where the treatment $X$ is randomly assigned to the subjects. In that case, $P(Y | do(X = x)) = P(Y | X = x)$, and the ATE can be estimated from the joint distribution of $X$ and $Y$. However, randomised controlled trials are expensive, and one commonly has to rely on observational data. In that case, the ATE is not directly observable, because $P(Y | do(X = x)) \neq P(Y | X = x)$. In fact, probabilities under interventions are *by definition* not derivable from the joint distribution only. One always needs additional information about the causal structure, usually given in the form of a directed acyclic graph (DAG) called the causal graph. To estimate the effect of interventions, Pearl [20] developed rules that relate interventional quantities on the causal graph to observational quantities relative to a transformed graph, collectively referred to as the do-calculus. By combining knowledge

of the joint distribution and the causal graph, one can estimate the effect of interventions, or conclude that an effect is not identifiable. It is these true causal effects, defined in terms of interventions, that our method decomposes.

Since we aim to capture the total causal power in a set of variables, not just the effect of a binary treatment, we slightly modify the ATE to a 'maximal average causal effect' by defining:

$$\text{MACE}(X_S; Y) = \max_{x,x' \in \mathcal{X}_S} \left( \mathbb{E}[Y|do(X_S = x)] - \mathbb{E}[Y|do(X_S = x')] \right) \tag{8}$$

where $\mathcal{X}_S$ is the set of possible values that $X_S$ can take. Note, however, that this is just one way to characterise causal strength. For instance, one could instead maximise the difference $\mathbb{E}[Y|do(X_S = x)] - \mathbb{E}[Y]$ with the observational state. Many definitions are possible (see e.g. [15] or the alternative explored in Appendix C), each of which can be similarly decomposed using our method. Further note that the MACE is not a measure of the causal effect of $X_S$ on $Y$, but rather a measure of the maximal causal power that $X_S$ can exert on $Y$. This is an important distinction, as it identifies which variables have causal power, but does not indicate which interventions will lead to which outcomes.

The MACE quantifies the maximal causal power that a set of variables $X_S$ can exert on $Y$. This definition has the advantage that it does not rely on *a priori* assumptions about the possible values of the variables, but it no longer captures the direction of the causal effect. It is this quantity that we wish to decompose into synergistic, redundant, and unique components. This makes sense, since while the MACE quantifies the total causal power of a set of variables, it does not tell us *how* this power is distributed among the variables. To understand and steer the behaviour of causal systems, it can be crucial to know if the causal power is redundant, synergistic, or unique.

### 3.2 Decomposing the causal effects into antichains

In order to decompose the MACE over the antichains, it is necessary to extend its domain to all antichains. For antichains of cardinality one, the definition from Equation (8) can be used. For a general antichain $\alpha$, we define this 'redundant' MACE, denoted as $\text{MACE}_\cap(\alpha; Y)$, as the minimum of the MACEs of elements of the antichain, since this is the causal power that they share:

$$\text{MACE}_\cap(\alpha; Y) = \min_{A \in \alpha} \text{MACE}(X_A; Y) \tag{9}$$

$$= \min_{A \in \alpha} \max_{x,x' \in \mathcal{X}_A} \left( \mathbb{E}[Y|do(X_A = x)] - \mathbb{E}[Y|do(X_A = x')] \right) \tag{10}$$

Note, however, that one could come up with different definitions of redundant causality, as long as it reduces to the chosen definition of the causal effects on antichains of cardinality one. With this definition, we can decompose the MACE of a set of variables $X_T \subseteq X_S$ over the lattice of antichains as

$$\text{MACE}(X_T; Y) = \text{MACE}_\cap(\{X_T\}; Y) = \sum_{\alpha \leq \{X_T\}} C(\alpha; Y) \tag{11}$$

where $C(\alpha; Y)$ denotes the 'partial causal effect' of antichain $\alpha$ to the full MACE. This corresponds to a 'partial causality decomposition'. A Möbius inversion then shows that the partial causal effects can be written as

$$C(\beta; Y) = \sum_{\alpha \leq \beta} \mu_{\mathcal{A}(X_S)}(\alpha, \beta) \text{MACE}_\cap(\alpha; Y) \tag{12}$$

where $\mu_{\mathcal{A}(X_S)}$ is the Möbius function of the antichain poset $(\mathcal{A}(X_S), \leq)$ which can be calculated using the fast Möbius transform [14]. One nice property of the $\text{MACE}_\cap$ is that it is a monotone function on the antichain poset:

**Lemma 1.** *The function* $\text{MACE}_\cap : \mathcal{A}_n \to \mathbb{R}$ *is monotonic with respect to the redundancy ordering on* $\mathcal{A}_n$.

*Proof.* See Appendix A. □

This lemma serves to lend credibility to our claim that the definition of $\text{MACE}_\cap$ is sensible, but also implies the following:

**Theorem 2.** *The partial causal effects* $C(\alpha; Y)$ *are nonnegative.*

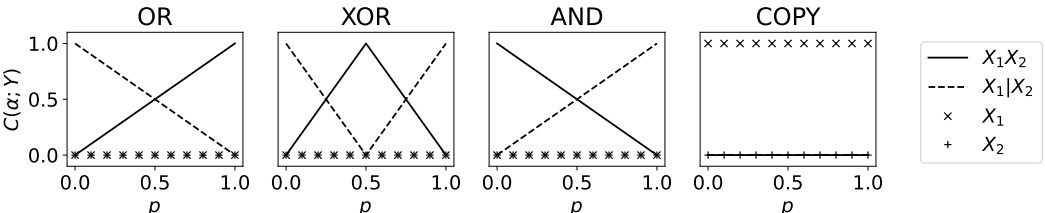

Figure 2: The causal contributions $C(\alpha; Y)$ of the antichains of logical gates.

*Proof.* Immediate by inserting Lemma 1 into the proof of Theorem 5 in [25]. □

Nonnegativity of causal effects makes intuitive and practical sense. Consider the following:

$$\text{syn}(S; Y) = \sum_{\substack{\alpha \in \mathcal{A}(S) \\ \nexists A \in \alpha : |A| = 1}} C(\alpha; Y) \tag{13}$$

This quantity $\text{syn}(S; Y)$ is the total aggregated synergistic causal power within the set of variables $S$ on $Y$. It captures all causal power that requires coordination among multiple variables. Nonnegativity of the $C(\alpha; Y)$ implies that even when not all these terms in the sum can be calculated, a partial sum will still give a lower bound on $\text{syn}(S; Y)$.

## 4 Decomposing causality in practice

Code to reproduce the figures in this section is available at [12].

### 4.1 Causal power in logic gates

The causal graph of a logic gate with two inputs is simply a collider structure: $X_1 \rightarrow Y \leftarrow X_2$. In this case, the do-operator reduces to the see-operator (by the back-door criterion $P(Y|do(X)) = P(Y|X)$ if $Y \perp\!\!\!\perp X$ in $G_{\underline{X}}$, the graph with all arrows out of $X$ removed), that is, $E(Y|do(X_1)) = E(Y|X_1)$, so we can just use the conditional expectation to calculate the MACE.

Let the two inputs be independently drawn from a binomial distribution with $P(X_1 = 1) = P(X_2 = 1) = p$, and the output be their logical OR, AND, XOR, or COPY, where $\text{COPY}(X_1, X_2) = X_1$. The MACE for each of these, as a function of $p$, can be easily calculated by hand. For example,

$$\text{MACE}_{\cap}(\{1\}; X_1 \text{ OR } X_2) = \mathbb{E}[X_1 \text{ OR } X_2|do(X_1 = 1)] - \mathbb{E}[X_1 \text{ OR } X_2|do(X_1 = 0)] \tag{14}$$
$$= 1 - p \tag{15}$$
$$\text{MACE}_{\cap}(\{1\}; X_1 \text{ XOR } X_2) = |1 - 2p| \tag{16}$$

The causal contributions $C(\alpha; Y)$ are then simply the Möbius inversion of these values, and shown in Figure 2. This shows that the causal contributions indeed disentangle the unique, redundant, and synergistic interventional power of the variables. In the OR gate, at low $p$ the causal power is very redundant: changing either output to a 1 will probably change the output state. In contrast, at high $p$ the only way to affect the outcome is generally to set both variables to 0, which requires synergistic causal control. The inverse is true for the AND gate. Controlling the XOR of the inputs at $p = \frac{1}{2}$ requires full synergistic control, whereas for large or small $p$ the causal power is mostly redundant because flipping either to 1 at low $p$, or to 0 at high $p$, tends to activate the output. The unique causal power of an input vanishes in all gates that are symmetric under permutations of the inputs. In contrast, the causality in the COPY gate is completely built from the unique causal power of $X_1$.

### 4.2 Causal power in cellular automata is context-dependent

A more interesting causal structure is that of a cellular automaton. Consider a 1D cellular automaton with $N$ cells, where each cell can be 0 or 1. The state of a cell at time $t + 1$ is determined by the state of itself and its two neighbours at time $t$. We consider the causal power a cell $B$ and its two neighbours at time $t$ have over $B$ at time $t + 1$. The causal graph takes the following form:

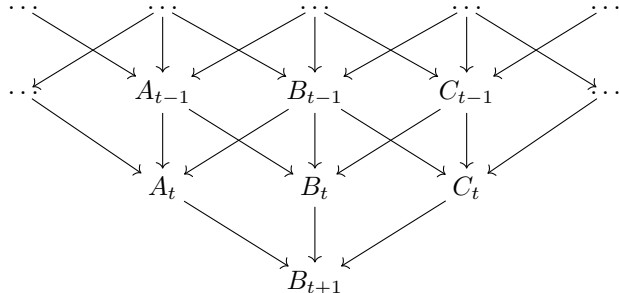

The MACE of $A$ can be written as

$$\text{MACE}(A_t; B_{t+1}) = \max_{s,s'} \left( \mathbb{E}[B_{t+1}|do(A_t = s)] - \mathbb{E}[B_{t+1}|do(A_t = s')] \right)$$

which shows that we need access to quantities like $p(B_{t+1}|do(A_t = s))$. Note that there are a lot of backdoor paths from $A_t$ to $B_{t+1}$, but all are blocked by the set $\{B_t, C_t\}$, so we can write:

$$p(B_{t+1}|do(A_t = a_t)) = \sum_{b_t, c_t} p(B_{t+1}|A = a_t, B_t = b_t, C_t = c_t)p(B_t = b_t, C_t = c_t)$$

$$p(B_{t+1}|do(A_t = a_t, C_t = c_t)) = \sum_{b_t} p(B_{t+1}|A = a_t, B_t = b_t, C_t = c_t)p(B_t = b_t)$$

$$p(B_{t+1}|do(A_t = a_t, B_t = b_t, C_t = c_t)) = p(B_{t+1}|A = a_t, B_t = b_t, C_t = c_t)$$

However, we do not have a clear prior probability distribution on $B_t$ and $C_t$, so there are multiple ways to evaluate these expressions. First, we consider the 'maximum entropy' solution, which simply assumes that the input states are drawn from a uniform distribution. For a single intervention, this implies:

$$p(B_{t+1}|do(A_t = s)) = \frac{1}{4} \sum_{b_t, c_t} p(B_{t+1}|A_t = s, B_t = b_t, C_t = c_t) \tag{17}$$

Decomposing causal effects under this assumption essentially decomposes the causal power under maximal ignorance. To calculate the MACE we just need to calculate the average effect of $A_t$ on $B_{t+1}$, conditional on the possible states of $B_t$ and $C_t$, which can be immediately read off from the rule specification. Another option is to let the inputs always be zero, so that $p(B_t, C_t) = \delta_{B_t,0}\delta_{C_t,0}$, which gives the causal power in the context of the empty state.

Alternatively, we could estimate the prior distribution of $B_t$ and $C_t$ based on data from a simulated automaton, which gives the causal power decomposition of the dynamics conditional on an initial state. We will consider two possible initialisations: a random initialisation, where each cell is drawn from a Bernoulli distribution with $p = 0.5$, and a middle-1 initialisation, where all cells are zero except for the middle cell, which is set to one.

The studied rules and their causal decompositions are shown in Figure 3, for each of the priors. To get empirical estimates, we simulated automata with 100 cells and periodic boundary conditions that evolved over 10k steps, where the first 500 states were discarded.

The results give a nuanced and context dependent description of the causal power inside the automata. For example, if you want to control the 'Rule 90' automata with a single intervention, then this is possible under a middle-1 initialisation, but not under a random initialisation. More details on the decomposition for each of the shown rules are available from Appendix B.

### 4.3 Rate-dependency in the causal decomposition of a chemical network

We are interested in seeing how the causal decomposition of a system can change as one varies a parameter. To illustrate this, consider the following chemical network. Two chemicals $X_1$ and $X_2$ can spontaneously form a molecule $Y$ at rates $k_1$ and $k_2$, but can also come together to form $Y$ at rate $k_3$. The molecule $Y$ then spontaneously degrades at rate $k_4$, which we set equal to 1. The concentration of $Y$ is described by the following rate equation and steady-state concentration:

$$\frac{d[Y]}{dt} = k_1[X_1] + k_2[X_2] + k_3[X_1][X_2] - k_4[Y] \tag{18}$$

$$[Y]_{ss}(X1, X2) = k_1[X_1] + k_2[X_2] + k_3[X_1][X_2] \tag{19}$$

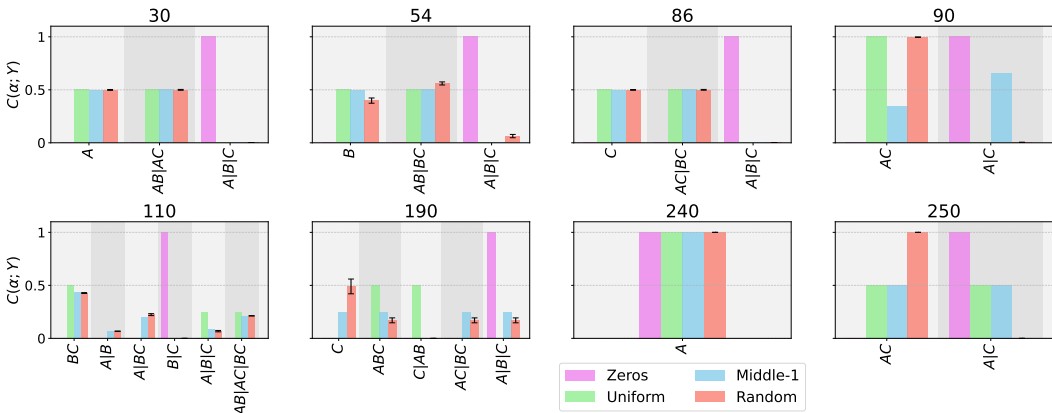

Figure 3: The causal decomposition of the cellular automata as a function of the probability of the inputs. Rule number indicated above each figure. Error bars indicate standard deviation across 20 random initialisations. Only antichains with a causal power of at least 0.01 in at least one context are shown.

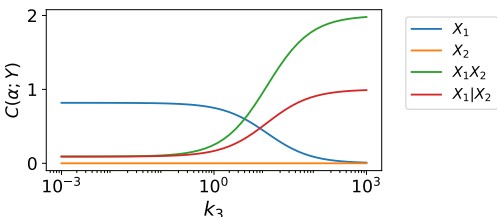

Figure 4: The causal power of perturbations in a chemical system. Parameters are set to $k_1 = 10$, $k_2 = 1$, $[X_1] = [X_2] = \epsilon = 1$.

We imagine that the experimenter is able to modify the concentrations of $X_1$ and $X_2$ by adding an amount $\delta_i$ to the concentration of $X_i$, and that the target variable is the normalised concentration $\hat{Y} = \frac{Y}{Y_{ss}[X_1, X_2]}$. Note that the causal structure of this system is still just a collider $X_1 \to Y \leftarrow X_2$, so to calculate the effect of an intervention on one of the concentrations, we just need to calculate the conditional expectation of $Y$ given the intervention, which is the steady state concentration (assuming that the system equilibrates quickly).

$$E(\hat{Y}|do(\delta_1 = \epsilon)) = \frac{[Y]_{ss}(X_1 + \epsilon, X_2)}{[Y]_{ss}(X_1, X_2)} \quad (20)$$

The MACE of an antichain of variables $\{X_1, X_2\}$ on $\hat{Y}$ is then again simply given by Equation (10), where the inner maximisation is trivial because we assume that the intervention is either of size $\epsilon$ or 0. The different causal contributions are shown in Figure 4 as a function of the rate $k_3$. At low $k_3$, spontaneous synthesis of $Y$ dominates, and since $k_1 > k_2$ almost all causal power lies uniquely with $X_1$. At high $k_3$, the combinatorial synthesis dominates, which is reflected by the fact that the causal power lies mostly with synergistic control. At high $k_3$, the asymmetry in spontaneous synthesis of $Y$ is no longer relevant, so all non-synergistic causal power is left redundant.

### 4.4 Synergistic and redundant semantics in a transformer language model

Finally, we demonstrate our causal decomposition in a more realistic scenario: we study the causal effect of string completions on sentiment analysis scores by the `distilbert-base-uncased-finetuned-sst-2-english` language model [23], as made available through the `Transformers` Python library [26]. Let the baseline sentence be "this movie is". Let an intervention correspond to appending a word to the baseline sentence, and define the

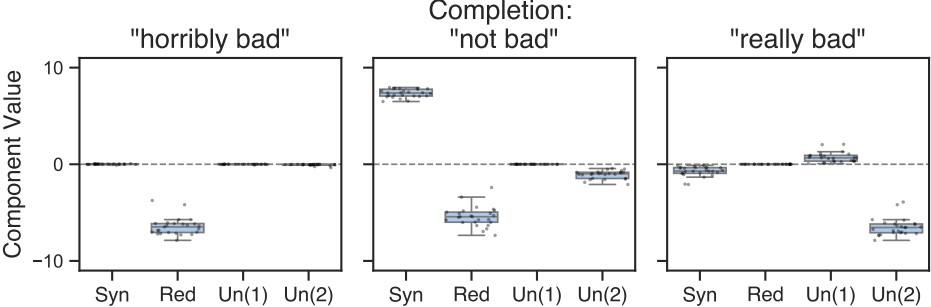

Figure 5: The causal decomposition of sentence completions in a language model for sentiment analysis reveals that semantic synergy, redundancy, and uniqueness are accurately captured. The box plots show the distribution of the causal components across 25 base sentences (see Appendix D).

causal effect of appending string A to be:

$$\mathrm{CE}(A; Y) = Y(\texttt{"this movie is"} + A) - Y(\texttt{"this movie is"}) \tag{21}$$

where $Y$ is the model's (logit) positivity score, and addition represents string concatenation. Since the causal effect is now signed, we slightly modify the definition of the redundant causal effects to preserve the sign:

$$\mathrm{CE}_\cap(\alpha; Y) = \begin{cases} \min_{A \in \alpha} \mathrm{CE}(A; Y) & \text{if } \forall A \in \alpha : \mathrm{CE}(A; Y) > 0 \\ \max_{A \in \alpha} \mathrm{CE}(A; Y) & \text{if } \forall A \in \alpha : \mathrm{CE}(A; Y) < 0 \\ 0 & \text{otherwise} \end{cases} \tag{22}$$

That is, the redundant causal effect is the strongest signed effect that all elements from $\alpha$ can achieve. With this, we can decompose the effect of sentence completions into synergistic, redundant, and unique effects of words. We investigated examples of each of these three components: the completion `"horribly bad"` represents redundant semantics (both words generally signal negative sentiment), `"not bad"` represents synergistic semantics (the combined meaning is different from that of the individual words), and `"really bad"` represents unique semantic content (the sentiment is fully contained in the word `"bad"`). The results are summarised in Figure 5. The decomposition matches with the above intuitions about synergistic, redundant, and unique semantic content, as the decomposition for `"horribly bad"` shows strong negative redundancy, `"not bad"` shows strong positive synergy that reflects the semantic negation (as well as a negative redundancy, because the model associated both `"not"` and `"bad"` with negative sentiment), and `"really bad"` shows strong unique negative sentiment for the word `"bad"`.

## 5  Discussion

We have presented a principled decomposition of causal effects into synergistic, redundant, and unique components. Decomposing causal effects in logic gates, cellular automata, chemical networks, and language models revealed the synergistic, redundant, and unique causal power of different variables. Furthermore, we illustrated how the causal decomposition can depend on the dynamical context of the system, or on the value of its parameters. These insights allow for a more nuanced understanding of the causal structure of a system, and can be used to guide interventions in a system to achieve more desirable outcomes.

The formalism is similar in spirit and algebra to that of the partial information decomposition, but by decomposing an interventional quantity we are able to disentangle synergistic and redundant causal power. The approach outlined here would in principle work for any quantity where a notion of synergy and redundancy can be defined, in particular for other definitions of causal power than the one in Equation (8). We encourage and anticipate further exploration of this approach, as our definition of the redundant MACE is deliberately simple to keep the presentation of the general formalism transparent. Two examples of extending the definition of redundant causality were already presented: one that preserves the sign of the effect of an intervention (Section 4.4), and one that replaces the

MACE by a counterfactual outcome (Appendix C). Both were inspired by the MMI measure from the PID literature, but one can imagine deriving different measures of redundant causality from other PID functions, or even completely novel ones. Causal decompositions in other contexts, like in climate systems with feedback loops, may require a more sophisticated notion of redundant causality. Janzing et al. [15], for example, define causal effects in terms of a Kullback-Leibler divergence, which can be similarly decomposed [13]. More generally, our approach fits into the wider context of deriving Möbius inverses of observable quantities to obtain a fine-grained description of complex systems [13], which might be used to derive different decompositions of causality in the future.

While the aim of this study is similar to that of Martínez-Sánchez et al. [19], we have taken a very different approach. We believe that decomposing causality fundamentally requires an interventional quantity, so decomposing a quantity derived purely in terms of the joint probability distribution, as is done by Martínez-Sánchez et al. [19] for mutual information, cannot yield true causal insight. While Martínez-Sánchez et al. [19] consider it an advantage of their method that it does not scale with the Dedekind numbers, we believe that the superexponential growth of the number of antichains is a feature, not a bug. It is a reflection of the fact that the number of ways information can be carried among $n$ variables fundamentally grows superexponentially with $n$. Martínez-Sánchez et al. [19] only consider redundancies among individual variables, not considering possible redundancies between pairs. For example, they do not include redundancies of the form $\{\{X_1, X_2\}, \{X_3\}\}$, or $\{\{X_1, X_2\}, \{X_3, X_4\}\}$, both of which are significantly nonzero in the cellular automata studied here. In general, their approach decomposes the total mutual information into $2(2^n - 1) + n$ terms. By not including the full set of redundancies, but still requiring that the sum of the terms adds up to the total, their method conflates multiple sources and does not disentangle the causal structure. While their results suggest that their method can be useful in understanding the structure of complex systems, we believe that it does not disentangle redundancy from synergy, and that they do not capture causal quantities.

**Limitations and future work** Our approach faces similar limitations to the partial information decomposition. Most pressingly, the number of ways in which causal relationships can be redundant and synergistic scales superexponentially with the number of variables. While this Dedekind scaling in theory limits our analysis to around five variables, we believe that in practice it does not strongly diminish the applicability of redundancy decompositions like the one introduced here. Our approach can yield very fast decompositions for up to five variables, the computational bottleneck being the calculation of the MACE, for which more efficient proxies might be found. While the fast Möbius transform from [14] can be employed to calculate parts of the decomposition on even larger systems, decompositions among more than five variables quickly become hard to interpret and are therefore not likely to be of practical relevance (partial causal effects quickly become hard to interpret, since the causal power among $n$ variables can be distributed among up to $\binom{n}{\lfloor n/2 \rfloor}$ sets by Sperner's Theorem).

A further limitation shared with the PID is the ambiguity of the definition of redundancy. We chose to base our definition of redundant causality on the (MMI) measure [1], which has been criticised for its simplicity [2]. Still, these limitations have not hindered widespread application of the PID to real-world systems (Luppi et al. [17] and Luppi et al. [18] being two recent MMI-based applications of the PID to human brain data), so similarly should not hinder real-world applicability of our approach.

Another limitation is causal inference itself: our method can decompose causal effects, but does not provide an easy way to estimate them. It is well-known that accurately estimating causal effects is a hard problem, which can limit the applicability of our method. Still, the decomposition can be used to gain insights in a variety of real-world systems where we can either do interventions, or have models that can simulate them. In systems with complex control, like gene regulation or the climate, there might be a combination of both redundant causality (which could improve robustness) and synergistic causality (which could allow for more efficient or fine-grained control). Understanding how to intervene in living systems to achieve a desired outcome is also a major challenge in synthetic biology and medicine. Disentangling causal power in AI systems is another important application, as it can help understand, steer, and align their behaviour. Lindsey et al. [16] recently developed a method to intervene on the computational graph of a large language model, which could be a very natural setting to apply our method. In the social sciences, one could imagine that individuals can have both redundant and synergistic control over the behaviour of the group. How to attribute responsibility, blame, or rewards in such a social network is both a quantitative and a philosophical/ethical question. We hope that our approach can provide some insight into the former.

## Acknowledgments and Disclosure of Funding

The author thanks Joris Mooij, Philip Boeken, Patrick Forré, Rick Quax, and Daan Mulder for helpful comments and suggestions concerning causality, synergy, and an early draft of this manuscript, and acknowledges support from the Dutch Institute for Emergent Phenomena (DIEP) cluster via the Foundations and Applications of Emergence (FAEME) programme.

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

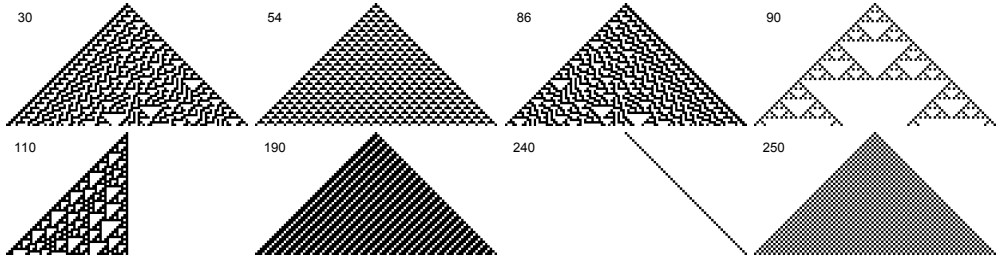

Figure 6: The various cellular automata rules studied, initialised with a single 1 in the middle of a 100-cell array and evolved for 50 steps.

## A    proofs

### A.1    Proof of Lemma 1

We first prove the following (obvious) fact:

**Lemma 2.** *The function* $\mathrm{MACE} : \mathcal{P}(X) \to \mathbb{R}$ *is monotonic on* $(\mathcal{P}(X), \subseteq)$.

*Proof.* Assume that $\alpha \subseteq \beta$. Let $a^*$ be the element of $\alpha$ that achieves the maximum, such that $\mathrm{MACE}_\cap(\alpha; Y) = \mathrm{MACE}(a^*; Y)$. Then, since $\alpha \subseteq \beta$, we know that $a^* \in \beta$, so $\mathrm{MACE}_\cap(\beta; Y) \geq \mathrm{MACE}(a^*; Y) = \mathrm{MACE}_\cap(\alpha; Y)$. Therefore, $\alpha \subseteq \beta \implies \mathrm{MACE}_\cap(\alpha; Y) \leq \mathrm{MACE}_\cap(\beta; Y)$. □

We can now prove Lemma 1:

**Lemma 1.** *The function* $\mathrm{MACE}_\cap : \mathcal{A}_n \to \mathbb{R}$ *is monotonic with respect to the redundancy ordering on* $\mathcal{A}_n$.

*Proof.* Recall that $\alpha \leq \beta \iff \forall b \in \beta, \exists a \in \alpha : a \subseteq b$. Now assume that given $\alpha \leq \beta$, we have

$$\mathrm{MACE}_\cap(\beta; Y) < \mathrm{MACE}_\cap(\alpha; Y) \tag{23}$$

$$\implies \min_{b \in \beta} \mathrm{MACE}(b; Y) < \min_{a \in \alpha} \mathrm{MACE}(a; Y) \tag{24}$$

By Lemma 2 Equation (24) can only be true as long as $\nexists b \in \beta, a \in \alpha : a \subseteq b$. However, since $\alpha \leq \beta$, we know that $\forall b \in \beta, \exists a \in \alpha : a \subseteq b$, which is a contradiction. Therefore, $\alpha \leq \beta \implies \mathrm{MACE}_\cap(\alpha; Y) \leq \mathrm{MACE}_\cap(\beta; Y)$. □

## B    Causal decomposition of cellular automata

The first few steps of a middle-1 initialisation of the automata rules are shown in Figure 6. We here provide further details on how the causal decompositions in Figure 3 relate to these automata rules.

Rule 30 can be written as $B_{t+1} = A_t$ XOR $(B_t$ OR $C_t)$, which shows that $A$ indeed has some unique causal power, but that there is also synergistic causality between $A$ and $C$. However, in a context of only zeros, the XOR reduces to a simple OR gate, which makes the causal power fully redundant.

Rule 54 corresponds to $B_{t+1} = (A_t$ OR $C_t)$ XOR $B_t$ so can be fully redundant in the context of zeros, but contains synergistic causality with $B$ in all other situations. Note, however, that there is redundancy among the two possible pairwise synergies with $B$.

Rule 86 is the mirrored version of rule 30, which is reflected by the equivariance of their decompositions under $A \leftrightarrow C$.

Rule 90 can be written as $B_{t+1} = A_t$ XOR $C_t$, which the causal decomposition correctly reveals as either purely synergistic or redundant causal power among $A$ and $C$, depending on the context. Only with middle-1 initialisation does the causal power become mixed, which is the initialisation that makes rule 90 evolve a fractal Sierpiński triangle.

Rule 110, famously complex and Turing complete, contains the most complex causal structure of the rules studied here, though this is not visible from the causal decomposition in the context of zeros.

Rule 190 can be written as $B_{t+1} = (A_t \text{ XOR } B_t) \text{ OR } C_t$, which is why the causal power is always fully redundant in the context of zeros, but shows both pure synergy as well as redundancy between $AB$ and $C$ in the uniform background. Under different initialisation, however, the causal decomposition become more complex.

Rule 240 is the exceptionally simple $B_{t+1} = A_t$, which is why every prior results in all causal power lying with $A$.

Rule 250 is $B_{t+1} = A_t \text{ OR } C_t$, and so, similar to the OR gate from Fig. 2 at $p = 0.5$, contains equal parts synergistic and redundant causal power, except in the context of zeros, or under random initialisation (which under this rule is equivalent to a context of only ones).

## C   Necessary and sufficient causes

To illustrate the flexibility of the framework for other causal quantities, consider decomposing the actual value of an outcome $Y$, which is the value that $Y$ takes when the input variables take their actually observed (binary) values $X_S = x_S$. We extend the definition of the outcome to antichains as:

$$Y_\cap(\alpha) = \min_{A \in \alpha} Y(do(X_A = 1), X_{S \setminus A} = x_{S \setminus A}) \tag{25}$$

This now describes a counterfactual quantity, in contrast to the MACE, which puts it on the same 'rung' as the usual notions of necessary and sufficient causes. We follow Halpern and Pearl [9], and define a cause to be a conjunction of primitive events. A Möbius inversion of $Y_\cap$ can identify necessary and sufficient causes as follows. We set $\forall i \in S : x_i = 0$ for simplicity, and calculate the inversion as

$$D(\beta; Y) = \sum_{\alpha \leq \beta} \mu_{\mathcal{A}(X_S)}(\alpha, \beta) Y_\cap(\alpha) \tag{26}$$

Similar as before, this yields a nonnegative decomposition when $Y_\cap$ is monotone on $(\mathcal{P}(X_S), \subseteq)$. When $D(\beta; Y) = 1$ then the conjunctions $\{\bigwedge_{b \in B} b \mid B \in \beta\}$ are the sufficient causes of $Y = 1$. A necessary cause is a cause that appears in every sufficient cause.

For example, when $Y(X_s) = \bigvee_{i \in S} X_i$ (cf. the disjunctive forest fire model from Halpern [8]), then $D(\beta; Y) = 1$ if and only if $\beta = \{\{X_i\} \mid i \in S\}$, indicating that activation of either of the variables is a sufficient cause of $Y$, but there are no necessary causes.

In contrast, when $Y(X_S) = \bigwedge_{i \in S} X_i$ (cf. the conjunctive forest fire model from Halpern [8]), then $D(\beta; Y) = 1$ if and only if $\beta = \{X_S\}$, indicating that the conjunction $\bigwedge_{i \in S} X_i = 1$ is both a necessary and sufficient cause of $Y = 1$. However, in a context where $x_i = 1$, $D(\beta; Y) = 1$ if and only if $\beta = \{X_{S \setminus \{i\}}\}$, indicating that a smaller conjunction is a sufficient cause.

When $Y(X_S) = 1$ iff $\sum_i X_i \geq 2$, then $D(\beta; Y) = 1$ if and only if $\beta = \{X_T \mid T \subseteq S, |T| = 2\}$, so conjunctions of two variables are sufficient causes of $Y = 1$.

As already noted by Halpern and Pearl [9], one might also allow causes to be formed from disjunctions. The set of all causes can then be ordered by logical implication, yielding the free distributive lattice generated by $X_S$. This happens to be isomorphic to the redundancy ordering on the antichains [14], so under this definition of causes, the decomposition simply assigns a value to every possible cause.

## D   Base sentences for sentiment analysis

The 25 base sentences prepended to the string completions in the sentiment analysis decomposition are:

```
"this movie is", "this book is", "I found this movie to be", "film rating:", "my
opinion of the film is that it's", "the acting was", "the plot felt", "overall,
I thought the picture was",  "the story is",  "this film is considered to be",
"I would describe this movie as",  "the director's work is",  "the script is",
"the new series is", "the novel is", "what I saw was", "the performance was",
"this show is", "the reviewer said it was", "my impression was that the film
is",  "the hotel room was", "my experience at the museum was",  "I found the
service", "Overall experience:", "In conclusion, the event was"
```

