# OpenReview forum: "Decomposing Interventional Causality into Synergistic, Redundant, and Unique Components"
_NeurIPS.cc/2025/Conference — NeurIPS 2025 spotlight_

### Official Review · Reviewer_Kx4Q · 2025-07-02

**Clarity:** 2
**Significance:** 3
**Originality:** 3
**Rating:** 5
**Confidence:** 3

**Summary:**

The authors propose a way to attribute the effect of an intervention in a causal Bayesian network into components that are associated with different sets. There decomposition is shows what part of the causal influence is synergetic among different variables, which part is redundant and which one can uniquely attributed to a single variable.

**Questions:**

-	The decomposition into synergetic, unique, and redundant information is only explicit for the example with two variables, the general case is only implicit, which I found hard to follow.
-	I hope I got the decomposition into antichains but missed how this defines a decomposition into synergetic, redundant, and unique information. Doesn’t the antichain decomposition amount to a decomposition into redundant components, each of which is synergetic? I don’t get it.
-	On a related note, I can roughly imagine what you mean by “A nice property of antichains is that they can be ordered with respect to redundancy by setting”, but I fail to understand it entirely.
-	Mention that poset stands for partially ordered set
-	Line 97: Y missing in I_delta
-	Figure 3 is suboptimal, the grouping could be more obvious. For instance, by dotted vertical lines that separate the antichains.
-	“The approach outlined here would in principle work for any macroscopic quantity, in particular for other definitions of causal power than the one in Equation (6)”. Why only macroscopic quantities? Maybe I just don’t understand the notion of macroscopic used here?
-	I don’t understand “Our definition of the redundant MACE is deliberatively naive to keep the presentation of the general formalism as transparent as possible”
-	“we believe that the superexponential growth of the number of antichains is a feature, not a bug”. This remark sounds like the mindset of pure theoreticians. Of course it is conceptually true, but it doesn’t help if one wants to apply it.
-	Reference [2] does not occur in the text
-	Reference [15] is incomplete
-	Which properties does an information measure need to satisfy to enable your decomposition?
-	wouldn’t the OR example in 4.1 better if X_1 and X_2 had different bias, that is, one has p_1 \neq p_2?

**Ethical Concerns:**

["NO or VERY MINOR ethics concerns only"]

**Final Justification:**

after crucial points clarified in the rebuttal, I can recommend the paper as solid work.

**Limitations:**

Practical applications are not obvious

**Quality:**

3

**Strengths And Weaknesses:**

Strength:
The decomposition described here is more principled that Shapley based approaches in the sense of being explicit about which part of the effect requires interactions, while Shapley averages over different background conditions.

The paper is very explicit about the distinction of association versus causality (rung 1 versus rung 2 in Pearl and Mackenzie’s ladder of causation).

Weakness:
The advantage of being more principled than Shapley does not come for free, it renders the decomposition cumbersome with a heavy combinatorial explosion of terms.

The explanations are sometimes lacking clarity since crucial parts are implicit, see below.

---

> ### Author Rebuttal · Authors · 2025-07-29
>
> Dear reviewer Kx4Q, thank you for your detailed review. We appreciate your honesty in flagging points of confusion and reflecting this in your confidence score.
>
> We in particular recognise that the relationship between synergy, redundancy, antichains, and their ordering can be confusing to those unfamiliar with the partial information decomposition (PID). Thank you for pointing this out—we hope that our response below can clear most of this up.
>
> Regarding your comments and questions:
>
> > W1: The advantage of being more principled than Shapley does not come for free, it renders the decomposition cumbersome with a heavy combinatorial explosion of terms.
>
> We appreciate you contrasting our method with Shapley values. It is true that Shapley values are averages over ‘coalition’ synergies. However, even if not averaged, these values still conflate redundancies and synergies because the coarser decomposition into sub coalitions does not distinguish the two.
>
> While this makes a combinatorial explosion inevitable, this has not hindered widespread adoption of the PID, which suffers from the same combinatorics (see also our response to Q9). Furthermore, the recently developed ‘Fast Möbius Transform’ has also enable a significant speedup of antichain decompositions, and reduced the complexity of some special components, like the pure synergy, to exponential (matching the complexity of Shapley value calculation) [1].
>
>
> > Q1. The decomposition into synergetic, unique, and redundant information is only explicit for the example with two variables, the general case is only implicit, which I found hard to follow.
>
> We have indeed only fully written out the 2-variable decomposition in the text, only highlighting 3 examples from the total of 18 terms in the 3-variable decomposition (in lines 105-106). We hope our answer to the next question clears up what the general case would look like.
>
> > Q2. I hope I got the decomposition into antichains but missed how this defines a decomposition into synergetic, redundant, and unique information. Doesn’t the antichain decomposition amount to a decomposition into redundant components, each of which is synergetic? I don’t get it.
>
> Your intuition is mostly correct: most antichains (namely, those that contain more than one set) indeed correspond to a redundant component of the decomposition. The remaining antichains—those that contain a single set—correspond to either unique of synergistic components. To summarise:
>
> - **Unique components** correspond to antichains with a single variable, e.g. $\\alpha = \\{\\{X_1\\}\\}$, and quantify how much of the causal power can be uniquely attributed to $X_1$.
>
> - **Synergistic components** correspond to antichains with a single set (of size >1) of variables, e.g. $\\alpha = \\{\\{X_1, X_2, …, X_n\\}\\}$, and quantify how much of the causal power is *only* accessible from joint interventions.
>
> - **Redundant components** correspond to antichains with multiple elements, like $\\alpha = \\{\\{X_1, X_2\\}, \\{X_3\\}\\}$,  and quantify how much of the causal power is accessible from *either* intervention.
>
> Redundancies of the type $\\{\\{X_1, X_2\\}, \\{X_3, X_4\\}\\}$ can, as you suggest, be considered as a ‘redundancy between synergies’, though this is mostly a linguistic choice. The important thing is its operational interpretation: it quantify how much of the causal power is accessible by a joint intervention on $X_1$ and $X_2$, **or** by a joint intervention on $X_3$ and $X_4$. Our method provides a value for *every* such mode of interaction, providing a complete decomposition into all possible redundant, synergistic, and unique components.
>
> > Q3. On a related note, I can roughly imagine what you mean by “A nice property of antichains is that they can be ordered with respect to redundancy by setting”, but I fail to understand it entirely.
>
> The ordering defined in equation (4) is indeed not trivial, and a key contribution of the original PID paper [2]. To see this, note that a component $\alpha$ is “more redundant” than $\beta$ if the requirements to access the information/causal power in $\beta$ are stricter than those you need for $\alpha$. For example, to access $\beta = \\{\\{X_1, X_2 \\}\\{ X_3, X_4\\}\\}$, you need to be able to do joint interventions on $X_1$ and$ X_2$ **or** on $X_3$ and $X_4$. In contrast, to access $\alpha = \\{\\{X_1, X_2 \\}\\{ X_3\\}\\}$, you need joint interventions on $X_1$ and $X_2$ **or** only a *single* intervention on $X_3$. Since $\alpha$ is thus 'easier' to access, we say it's more redundant and write $\alpha < \beta$. Note that *more* redundant corresponds to being ‘lower’ in the ordering (the standard in the PID literature). We will make sure to explain this more clearly in the camera-ready version. Note that while the ordering is crucial to carry out the calculation of the decomposition, it is not very important for conceptual understanding of the decomposition.
>
> > Q4. Mention that poset stands for partially ordered set
>
> Thank you---this is now fixed.
>
> > Q5. Line 97: Y missing in I_delta
>
> Not only was there a missing $Y$, there was also some notational inconsistency for $I_\\partial$ and $Y_\\cap$ in this paragraph. This is now fixed.
>
> > Q6. Figure 3 is suboptimal, the grouping could be more obvious. For instance, by dotted vertical lines that separate the antichains.
>
> We have shaded the background behind every other group to clearly visually separate them. This has made the figure much clearer. Note that in response to reviewer BJo1, we have also increased the font size in this figure to make everything more legible.
>
> > Q7. “The approach outlined here would in principle work for any macroscopic quantity, in particular for other definitions of causal power than the one in Equation (6)”. Why only macroscopic quantities? Maybe I just don’t understand the notion of macroscopic used here?
>
> We recognise and agree that the use of ‘macroscopic’ here does not aid clarity. It is borrowed from the literature on Möbius inversions more generally, but offers no insight in this case. We have changed the sentence to read “[...] would in principle work for any quantity where a notion of synergy and redundancy can be defined, in particular for of […]”.
>
>
> > Q8. I don’t understand “Our definition of the redundant MACE is deliberatively naive to keep the presentation of the general formalism as transparent as possible”
>
> We refer to our definition as naive because many applications will require more sophisticated measures of causality---ones that keep track of signed effects, possible outcomes, etc. An example of this is shown in the discussion of sufficient and necessary causes in Appendix C. Another example is the definition introduced in the context of semantic synergy in response to a comment by reviewer EPS7. However, to keep the main text as simple as possible, we decided to centre the main text around a straightforward and somewhat crude or ‘naive’ measure.
>
>
> > Q9. “we believe that the superexponential growth of the number of antichains is a feature, not a bug”. This remark sounds like the mindset of pure theoreticians. Of course it is conceptually true, but it doesn’t help if one wants to apply it.
>
> We respectfully disagree: we consider the rich algebraic structure a 'feature', because it can reveal very complex and nuanced causal dynamics in even small systems. It is simply a fact that there is a superexponential number of ways in which causal power can be distributed among variables. Scientific applications of the PID on real data sets show that the combinatorial explosion does not make practical applications intractable (see e.g. [3, 4]).
>
> > Q10. Reference [2] does not occur in the text
>
> While reference [2] is not discussed in detail, it is referenced as one of the studies that introduced a redundancy measure in the parentheses on line 122.
>
> > Q11. Reference [15] is incomplete
>
> Thank you for pointing this out. We have fixed this to read “Dominik Janzing, David Balduzzi, Moritz Grosse-Wentrup, and Bernhard Schölkopf. Quantifying causal influences. The Annals of Statistics, 41(5):2324–2358, 2013”
>
> > Q12. Which properties does an information measure need to satisfy to enable your decomposition?
>
> Whenever extending the definition of a measure of information or causality from sets of variables to antichains of variables is possible, then the decomposition applies. This means that one can apply this framework in any setting where a notion of ‘redundancy’ or ‘synergy’ can be defined. This is what allowed insights from the PID to be used to define a decomposition of causality.
>
> > Q13. wouldn’t the OR example in 4.1 better if X_1 and X_2 had different bias, that is, one has p_1 \neq p_2?
>
> One could certainly perform this analysis. If we assume w.l.o.g. that $p_1>p_2$, then the redundancy  $C(\\{1\\}\\{2\\}; X_1 \\text{ OR } X_2)= 1 - p_1$ and the synergy $C(\\{1, 2\\}; X_1 \\text{ OR } X_2)=p_2$. We believe this adds little new insight while requiring more plots. The main advantage would be showing that $X_1$ now gains a unique component, but this is already demonstrated with the COPY gate. Our aim with these first examples is to familiarise the reader with the decomposition and to illustrate that it matches with intuition, which is why we chose to keep the example as simple as possible.
>
>
> [1] Jansma, Abel, Pedro AM Mediano, and Fernando E. Rosas. "Fast Möbius transform: An algebraic approach to information decomposition." Physical Review Research 7.3 (2025): 033049.
>
> [2] Williams, Paul L., and Randall D. Beer. "Nonnegative decomposition of multivariate information." arXiv preprint arXiv:1004.2515 (2010).
>
> [3] Luppi, Andrea I., et al. "A synergistic core for human brain evolution and cognition." Nature neuroscience 25.6 (2022): 771-782.
>
> [4] Luppi, Andrea I., et al. "A synergistic workspace for human consciousness revealed by integrated information decomposition." Elife 12 (2024): RP88173.

---

> > ### Comment · Reviewer_Kx4Q · 2025-08-04
> > **answers convincing**
> >
> > I am fine with the answers and raise my score.

---

> > > ### Author Response · Authors · 2025-08-05
> > >
> > > Dear reviewer Kx4Q,
> > >
> > > Thank you sincerely for the time and effort you have dedicated to reviewing our paper, and for your timely response to our rebuttal. Your comments have substantially improved the paper.

---

### Official Review · Reviewer_EPS7 · 2025-07-03

**Clarity:** 3
**Significance:** 2
**Originality:** 2
**Rating:** 5
**Confidence:** 3

**Summary:**

This paper proposes a way to decompose the causal effect of a set of variables $X$ on $Y$ into different components: contributions from the individual variables in $X$, contributions arising only from controlling multiple variables jointly ("synergistic"), and contributions that could be achieved by intervening on several subsets of variables ("redundant"). Unlike earlier work with the same goal, this paper is based on the interventional notion of causality. Section 4 illustrates the behaviour of these measures on different simulation examples.

**Questions:**

I have just some minor suggestions:
- line 97: "that carried" should be "that is carried"
- line 175: the reference to Eq. (8) seems wrong; should this refer to the equation at the top of the page?
- the plots in Figure 2 should be labelled with the names of the gates. Right now, their order is OR - XOR - AND - COPY, which is not the order the reader expects.
- line 243: something went wrong here
- line 282: "deliberatively" should be "deliberately"

**Ethical Concerns:**

["NO or VERY MINOR ethics concerns only"]

**Final Justification:**

I'd like to thank the authors for their clear and extensive answer to my follow-up question. Together with the other discussions on this page, it has convinced me that this work is a substantial addition to what is currently possible with PID, and that the new sentiment analysis example is a good illustration of this.

There are still choices that a user of this methodology will need to make, including in particular how to measure the causal effect of a set of variables, and how to generalize this to antichains. There a many possibilities for these, but I believe the authors have made a good choice of sensible options to present here. I think it is justified to leave further exploration to future work. I have updated my rating accordingly.

**Quality:**

3

**Strengths And Weaknesses:**

I find the work interesting, and see that it is potentially useful to be able to decompose causal effects as described.

The technical quality of the paper looks solid, and the writing and structure are clear.

The originality of the work is limited, as the PID it is based on has already been applied in similar settings.

The three examples in section 4 all consider cases where all X's are causes of Y. Latent confounding only arises in example 2, the cellular automata. However, this example is still quite artificial. While it is good to have an artificial example (like example 1, the logic gates) to demonstrate how the decomposition behaves, for purposes of establishing the practical utility of the decomposition it would be good to include a more realistic example.

---

> ### Author Rebuttal · Authors · 2025-07-29
>
> Dear reviewer EPS7, thank you for your insightful review and honest evaluation. We particularly appreciate your characterisation of our approach as interesting, potentially useful, technically solid, and clear. We are also grateful for your suggestion to study a more realistic scenario, which we have done by studying semantic synergy in sentiment analysis models, as outlined below. We believe this indeed demonstrates the flexibility and utility of the framework more convincingly.
>
> Regarding your comments and questions:
>
> >W1. The originality of the work is limited, as the PID it is based on has already been applied in similar settings.
>
> We fully acknowledge that our work directly builds on the established PID framework. However, we believe our main contribution presents a novel method and a crucial conceptual shift: moving from decomposing an observational quantity (mutual information) to an interventional one (causal effects). Even in simple settings, we believe observational and interventional quantities should be clearly distinguished, and represent very different things. The ability to decompose causal quantities fundamentally changes what questions can be asked, and what insights can be gained.
>
> > W2. The three examples in section 4 all consider cases where all X's are causes of Y. Latent confounding only arises in example 2, the cellular automata. However, this example is still quite artificial. While it is good to have an artificial example (like example 1, the logic gates) to demonstrate how the decomposition behaves, for purposes of establishing the practical utility of the decomposition it would be good to include a more realistic example.
>
> The simplicity of our examples is intentional; they were chosen to serve as clear, intuitively verifiable proofs-of-concept for the method. That being said, we recognise the additional benefit of showcasing the method on more realistic data, and thank you for suggesting this.
>
> To act on this, but keep the results intuitively verifiable, we have used the method to detect semantic synergy and redundancy in a sentiment analysis pipeline. As we are not allowed to share links to code or figures here, we briefly describe what we did, and summarise results in a markdown table.
>
> We analysed the causal effect of string completions on sentiment analysis scores by the `distilbert-base-uncased-finetuned-sst-2-english` model. Let the baseline sentence be “this movie is”. Let an intervention correspond to appending a word to the baseline sentence, and define the causal effect of appending string A to be:
>
> $$
> CE(A;Y) = Y(\text{"this movie is"} + A) - Y(\text{"this movie is"})
> $$
>
> where addition represents string concatenation, and $Y$ represents the (logit) positivity score from the model. Since the causal effect is now signed, we define the redundant causal effect as
>
> $$
> CE_\\cap(\\alpha;Y) = \\begin{cases}
> \\min_{A \\in \\alpha} CE(A;Y) \\quad \\text{ if }  \\forall A\\in \\alpha: CE(A;Y)>0 \\\\
> \\max_{A \\in \\alpha} CE(A;Y) \\quad \\text{ if }  \\forall A\\in \\alpha: CE(A;Y)<0 \\\\
> 0 \\text{ else}
> \\end{cases}
> $$
>
> That is, the redundant causal effect is the strongest signed effect that all elements from $\\alpha$ can achieve. With this, we can decompose the effect of sentence completions into synergistic, redundant, and unique effects of words. We investigated examples of each of these three kinds. The results are summarised in a table below, and make intuitive sense:
>
> **Redundant semantics:** An example of a mostly redundant completion would be “this movie is horribly bad”—both words are negative. This is correctly reflected by the decomposition: Only the redundant effect between “horribly” and “bad" is significantly nonzero (the redundancy is -7.13, reflecting a strong negative redundancy, whereas the synergy and unique components are all <0.03 in absolute value).
>
> **Synergistic semantics:** An example of a mostly synergistic completion would be “this movie is not bad”. Both “not” and “bad” by themselves decrease positivity in the model, but their combination is strongly positive. This is reflected by the decomposition: A synergy of 7.8, reflecting a strong positive synergistic effect. The significant redundancy reflects the negativity inherent in both words. The unique effects are much smaller.
>
> **Unique semantics:** An example of a completion where the full semantics are uniquely captured by a single word is “this movie is really bad”. The decomposition reveals that the sentiment is almost fully the result of the word “bad”: all components are smaller than 1 in absolute value, except the unique contribution of “bad”, which is strongly negative.
>
> Summary:
>
> | Text completion              | synergy   | redundancy | unique 1 | unique 2 |
> |-----------------------------|-----------|------------|----------|----------|
> | “horribly bad”              | 0.02      | **-7.13**  | 0.00     | -0.02    |
> | “not bad”                   | **7.83**  | -6.32      | 0.00     | -0.83    |
> | “really bad”                | -0.62     | 0.00       | 0.62 | **-7.15**    |
>
>
>
>
>
> We believe this example, while still just a proof-of-concept, demonstrates the practical utility of the method in a more realistic setting, and we will add it to the paper. For the camera-ready version, we will put error bars on all these values by decomposing the completion relative to a range of baseline sentences (“this movie is”, “this book is”, “I found this movie to be”, “film rating:”, etc.), to attenuate context sensitivity (our preliminary checks showed all results to reproduce across these contexts).
>
> We will also add to the discussion how the framework could be applied to more complex systems. For example, in line with suggestions by reviewers dQV8, we will discuss how one could use the framework to disentangle causal contributions in the attribution graphs of LLMs [1].
>
>
> >Q1. line 97: "that carried" should be "that is carried”
>
> Fixed.
>
> >Q2. line 175: the reference to Eq. (8) seems wrong; should this refer to the equation at the top of the page?
>
> Correct, this was a referencing issue that is now fixed.
>
> >Q3. the plots in Figure 2 should be labelled with the names of the gates. Right now, their order is OR - XOR - AND - COPY, which is not the order the reader expects.
>
> Fixed, the plots are now labelled.
>
> >Q4. line 243: something went wrong here
>
> Correct, this is now fixed and reads “The studied rules and their causal decompositions are shown in Figure 3, for each of the priors.”
>
> >Q5. line 282: "deliberatively" should be “deliberately"
>
> Fixed.
>
> > Limitations: no
>
> Judging by the reviewer guidelines, this response seems to signal that the reviewer could not find a discussion on the limitations of our method. We appreciate you raising this concern, and sincerely apologise that the discussion of limitations was not sufficiently clear. We tried to emphasise limitations of the method in the Discussion, specifically regarding our definition of redundant MACE (sentence starting on line 282) and the computational challenges posed by the superexponential growth of antichains (sentence starting on line 304). We apologise if this was not sufficiently highlighted and will add a dedicated and properly signposted "limitations" subsection to the discussion.
>
> [1] Ameisen, et al., "Circuit Tracing: Revealing Computational Graphs in Language Models", Transformer Circuits, 2025.

---

> > ### Comment · Reviewer_EPS7 · 2025-08-06
> > **Follow-up questions**
> >
> > Dear authors, thank you for your reply and your effort in addressing my concerns.
> >
> > About your reply to W2, I have two further question: 1) Is this definition of $CE_\cap(\alpha;Y)$ based on existing literature? 2) What aspects of this analysis would be different if they were carried out using the original, non-causal version of the PID?

---

> > > ### Author Response · Authors · 2025-08-08
> > >
> > > Dear Reviewer EPS7,
> > >
> > > Thank you sincerely for the time and effort you have dedicated to reviewing our paper, and for the opportunity to clarify our work in response to your questions. As the end of the discussion period is now less than 24 hours away, we wanted to make sure that we have adequately addressed your questions and concerns. Please do let us know if there are any remaining issues we can clarify or address.

---

> ### Author Response · Authors · 2025-08-06
> **Response to Q1**
>
> Dear reviewer EPS7,
>
> Thank you for your response. We will address your two questions in two comments.
>
> > Q1. Is this definition of based on existing literature?
>
> To our best knowledge, the above definition of $CE_\\cap(\\alpha;Y)$ has not been studied before, but it is not a new *ad-hoc* measure. While it looks more elaborate, it is just a signed version of the MMI-inspired redundancy measure for the MACE in equation (8) of the paper. In that sense, it was also inspired by the MMI from [1].
>
> The MMI approach to redundancy is capturing the "minimum" shared influence. Our signed measure $CE_\\cap(\\alpha;Y)$ adapts this principle to a context with positive and negative effects by minimising absolute value, but preserving the sign. This adaptation is necessary to handle semantic negation. For that reason, we need to add 3 clauses to the definition, where previously we only needed 1, though each clause reduces to "the smallest shared causal effect".

---

> > ### Author Response · Authors · 2025-08-06
> > **Response to Q2**
> >
> > > Q2. What aspects of this analysis would be different if they were carried out using the original, non-causal version of the PID?
> >
> > Our added analysis concerns the question *“how do words affect a model’s sentiment score for a text synergistically, redundantly, and uniquely”*. This question cannot properly be addressed with the PID for two conceptual reasons, and two practical ones.
> >
> > **Conceptual problem 1 (observational vs interventional):**
> > The question is interventional, whereas the PID only deals with correlations, which are observational quantities. This means that the PID can never truly answer interventional questions, but also leads to the problem of confounding (see practical problem 2 below).
> >
> > **Conceptual problem 2 (non-negativity of mutual information):**
> > Mutual information on two variables is non-negative so $I(A, Y(A))$ cannot distinguish between positive and negative effects of $A$ on the sentiment. Synergistic semantic negation, where a combination of words negates their separate meanings (e.g. “not bad”), is therefore not detectable by looking at mutual information.
> >
> > Still, one could imagine that $I(A, Y(A))$, the mutual information between a string $A$ and the model’s evaluation, can serve as a proxy for the effect of an intervention (though this requires a lot of assumptions about the underlying causal structure). This, however, still leaves some practical problems.
> >
> > **Practical problem 1 (estimating mutual information):**
> > To be precise: $A$ is the string to append to the baseline sentence “this movie is”, and $Y(A)$ is the model’s evaluation of the string “this movie is {A}”. To calculate $I(A, Y(A))$, we need to treat $A$ and $Y(A)$ as random variables with a joint distribution. The variable $A$ is a binary random variable that indicates whether a piece of text contains the string $A$. The variable $Y(A)$ is a random variable that describes the model’s sentiment score of a piece of text. We obviously do not have access to this joint distribution, so would have to estimate it from a large corpus of relevant text.
> >
> > On that large corpus of text, we’d need to keep track of how often “this movie is” is immediately followed by “not bad”, as well as all cases in which it is not followed by that. In addition, we need to evaluate the model on each piece of the text. But there are many more choices to make. For example: how do we define a ‘unit of text’. Say the following sentence appears: “this movie is not bad in terms of writing, but the acting is so bad that it ruins the whole film”. Should we count this as an occurrence of a negative sentiment that still contains the string “this movie is not bad”?
> >
> >
> > **Practical problem 2 (confounding):**
> > As you try to estimate the joint distribution from your chosen units of text, the structure of language introduces a strong effect of confounding: words like ‘bad’ and ‘not’ might occur more often in texts with negative sentiment, *even if the negative sentiment is largely caused by something else*.
> >
> >
> > Note that this does not even give us a decomposition of $I(\\alpha, Y(A))$, for which we would need to do the above procedure again for every element of $\alpha$. Even if we would be able to do this, it would be expensive, complicated, and ambiguous. To our best knowledge, there is no clear answer on how to solve these issues and perform a similar analysis with the PID.
> >
> >
> > Perhaps the difference can be best summarised as follows: when you are unable to do interventions, you can only decompose observational quantities like the mutual information (which can already be very interesting!). But observational quantities generally require just that: many observations. If you are able to actually intervene, then that is usually a much more efficient, direct, and unambiguous way to disentangle relationships between variables.
> >
> > This is especially true in real-world systems with many variables and complex interactions, like language, which inspired us to use this example to address your request for a more realistic example.
> >
> > We hope this clarifies our approach, and we thank you again for your engagement with our work. Please do let us know if you have any further questions.
> >
> > [1] Adam B Barrett. Exploration of synergistic and redundant information sharing in static and dynamical gaussian systems. Physical Review E, 91(5):052802, 2015.

---

### Official Review · Reviewer_dQV8 · 2025-07-22

**Clarity:** 4
**Significance:** 3
**Originality:** 4
**Rating:** 5
**Confidence:** 4

**Summary:**

The paper extends work in Partial Information Decomposition to the problem of decomposing causal effects under intervention. It takes advantage of Möbius inversion to provide a method for decomposing causal effects in terms of synergistic, redundant our unique power of interventions on causal variables.

The paper shows that this decomposition can elucidate the causal power of interventions in a set of simple but increasingly realistic causal systems.

**Questions:**

Is a combination of the purely intervention based notion of causal analysis here and the approach in the Martínez-Sánchez et al paper possible? Would such a combination allow for speedups.

Can you think of clear paths towards using this technique to extend the internal circuit tracing analysis in Jack Lindsey, Wes Gurnee, Emmanuel Ameisen, Brian Chen, Adam Pearce, Nicholas L. Turner, Craig Citro, David Abrahams, Shan Carter, Basil Hosmer, Jonathan Marcus, Michael Sklar, Adly Templeton, Trenton Bricken, Callum McDougall, Hoagy Cunningham, Thomas Henighan, Adam Jermyn, Andy Jones, Andrew Persic, Zhenyi Qi, T. Ben Thompson, Sam Zimmerman, Kelley Rivoire, Thomas Conerly, Chris Olah & Joshua Batson (2025).
“On the Biology of a Large Language Model.” https://transformer-circuits.pub/2025/attribution-graphs/biology.html

**Ethical Concerns:**

["NO or VERY MINOR ethics concerns only"]

**Final Justification:**

The authors have given thoughtful responses to the reviewers, including revisions.  which only reinforces my view that the paper should be accepted. I have reviewed all of the author responses to my and other reviews, and believe that the paper should be accepted.  Significant small issues with clarity of exposition have been rectified, and additional experiments have been done to ground the theoretical work in a straightforward sentiment analysus setting.

**Limitations:**

Yes

**Paper Formatting Concerns:**

No concerns

**Quality:**

3

**Strengths And Weaknesses:**

The paper provides an excellent exposition of the relevant parts of PID and causal inference, and introduces a technique that can take advantage of the newly discovered fast Möbius transform. Although the available speedup is not sufficient at this point to improve the causal analysis of complex systems (> 5 variables) it provides a foothold for approximation techniques that might lead to analysis of large systems (including, perhaps, the causal structure of LLMs).

There is a small typo in the definition of the Möbius function:
$\mu_S : S \times S \to \mathbb{R}$, should be $\mu_S : S \times S \to \mathbb{Z}$

---

> ### Author Rebuttal · Authors · 2025-07-29
>
> Dear reviewer dQV8, thank you for your insightful review and positive but honest evaluation.
>
> Regarding your comments and questions:
>
> > Is a combination of the purely intervention based notion of causal analysis here and the approach in the Martínez-Sánchez et al paper possible? Would such a combination allow for speedups.
>
> Yes, this is a very interesting possibility. Martínez-Sánchez et al never actually construct the partial order on which their decomposition is based, so they cannot exploit the fact that for a fixed partial order structure their decompositions can be reduced to a computationally efficient matrix-vector multiplication (the matrix being a representation of the Möbius function), as was done in [1]. If one manages to calculate the Möbius function on this partial order, this could modify and potentially drastically speed up their method to decompose the mutual information.
>
> > Can you think of clear paths towards using this technique to extend the internal circuit tracing analysis in Jack Lindsey [et al]
>
> Yes. In their report, Jack Lindsey et al note that interventions on features in the attribution graph have causal effects on the output probabilities of certain tokens. We can treat the attribution graph as a causal graph (it’s a DAG), the probability of the desired output token as the target variable, and intervene on other features in the graph. This will reveal exactly how the features interact redundantly vs synergistically to predict the output token. This is an important application we envision, though constructing and intervening on attribution graphs is itself not trivial.
>
> > Typo
>
> Thank you for catching the typo regarding the Möbius function's codomain; you are correct that it should be $\mathbb{Z}$. This is now fixed.
>
> [1] Jansma, Abel, Pedro AM Mediano, and Fernando E. Rosas. "Fast Möbius transform: An algebraic approach to information decomposition." Physical Review Research 7.3 (2025): 033049.

---

### Note · Authors · 2025-08-15

We sincerely thank the AC and all reviewers for their thoughtful engagement and constructive feedback. This review process has been very valuable, and has substantially improved our manuscript.

The discussions prompted us to demonstrate the practical utility of our framework in a more realistic setting, a key point raised by reviewer EPS7. In response, we introduced a new experiment analysing semantic synergy and redundancy in a sentiment analysis model. We validated this new result across 25 base sentences and 3 sentence completions, and have includes a figure with boxplots in the camera-ready version that illustrate the results are fully reproducible. We believe this new application, along with our other examples, provides a compelling and well-rounded validation of our method's contribution.

We are grateful for the strong support from our reviewers. Reviewer dQV8 recommended acceptance from the start, and we were pleased that our detailed clarifications convinced Reviewer Kx4Q, who found our answers "convincing" and subsequently raised their score.

We also appreciate the insightful follow-up questions from Reviewer EPS7, which allowed us to further clarify the novelty and practical advantages of our interventional approach compared to observational methods.

We are confident that we have fully addressed all concerns.

Changes we have implemented for the camera-ready version:

- Added the new experiment (in response to reviewer EPS7) to the results section.
- Added a section in the discussion on using other redundancy functions, beyond MMI inspired ones.
- Mentioned the suggested LLM application in the discussion (as suggested by reviewer dQV8).
- Added a clearly signposted ‘Limitations’ section to the Discussion.
- A short explanation of the redundancy ordering (as requested by Kx4Q).
- Fixed typos and formatting of text, equations, and figures, as flagged by the reviewers.

---

### Decision · Program_Chairs · 2025-09-17

**Decision:**

Accept (spotlight)

**Comment:**

The paper extends work in Partial Information Decomposition to the problem of decomposing causal effects under intervention. It takes advantage of Möbius inversion to provide a method for decomposing causal effects in terms of synergistic, redundant our unique power of interventions on causal variables. The paper shows that this decomposition can elucidate the causal power of interventions in a set of simple but increasingly realistic causal systems. The proposed decomposition is more principled that Shapley based approaches in the sense of being explicit about which part of the effect requires interactions. The formalism is also very explicit about the distinction of association versus causality.
Because of these strengths, the paper is deemed of great interest for the NeurIPS community.